# Biogeochemistry of Dominant Plants and Soils in Shewushan Gold Lateritic Deposit, China

**DOI:** 10.3390/plants11010038

**Published:** 2021-12-23

**Authors:** Haoyang Qin, Zhenghai Wang

**Affiliations:** School of Earth Science and Engineering, Sun Yat-Sen University, Guangzhou 510275, China; qinhaoy@mail2.sysu.edu.cn

**Keywords:** plant response, soil, abiotic stress, biogeochemistry

## Abstract

This paper describes the effect of mineral elements on dominant plants in the Shewushan lateritic gold deposit, China. For this purpose, 30 soil profile samples at different depths and 3 kinds of dominant plants including *Populus canadensis* (*Populus X canadensis Moench*), *Cinnamomun camphora* (*Cinnamomum camphora* (L.) *Presl.*) and Rhus chinensis (Rhus chinensis Mill.) were collected. The concentration of ore-forming elements including Au, Ag, Pb, Zn, Cu, As, Fe, and S were analyzed. Based on the investigation of two mine profiles, it can be found that Au, Pb, As, and Fe were mainly enriched in laterite layer and the brown clay layer at a depth of 5–11 m. Moreover, the biological accumulate coefficient (BAC) and the contrast coefficient (CM) were calculated to assess the sensitivity and concentrating ability of Populus canadensis and *Cinnamomun camphora*. To investigate the response of the two species to metal stress, the contents of chlorophyll, malondialdehyde (MDA), and activities of superoxide dismutase (SOD) and peroxidase (POD) were determined. The result showed that Populus canadensis and *Cinnamomun camphora* have a high tolerance to metal stress and that both of the two species can indicate the content of Au, As, Pb, and Co in topsoil.

## 1. Introduction

The development of plants is closely related to the biochemical quality of the soil environment quality. Studies have shown that some fern species such as *Dicranopteris dichotoma (Thunb.)Bernh,*
*Eucalyptus, Pityrogramma calomelanos* and *Pteris vittata* L. growing in surface mine areas can accumulate several kinds of metals [1,2,3,4,5]. These conclusions were reached by studying hundreds of dominant plants in mines. The stability of these plants’ physiology and their tolerance to heavy metal stresses are closely related to a more efficient antioxidant and protective enzyme system. Trace elements in the soil cause a series of biological reactions in plants, which can usually be observed through the changes in their enzyme activity in plant. Therefore, the growth state of dominant plants can indicate the content of heavy metal elements in the soil, the rhizosphere soil conditions, and the groundwater environment in a mining area [6,7]. Characterizing the heavy metal contents in leaves and the biochemical parameters of the plants could indicate the subsurface mineral resources, which is of great significance for mineral exploration and heavy metal pollution control [8,9,10].

The Shewushan gold mine is the largest laterite gold deposit in Asia, and the gold mainly occurs in the brown clay and the lateritic soil that had undergone weathering and leaching [11]. The reticulate lateritic and brown clay in the mining area is at 4–12 m depth with different thicknesses and high gold content. The dominant tree species in the mine area were *Populus canadensis* and *Cinnamomun camphora*. Studies have shown that the roots of these three kinds of dominant plants can grow to 5–12 m [12], and the roots from the soil profile in the lateritic soil layer can be found to have a high gold content [13]. In this paper, dominant plants including *Populus canadensis* and *Cinnamomun camphora* were selected to indicate the gold content in the study area. The distribution characteristics of gold and related elements were studied and analyzed through systematic sampling of the topsoil, the lateritic layer, and brown clay layer in the mining area. In addition, the heavy metal content and enzyme activity of the *Populus canadensis*, *Cinnamomun camphora,* in the mining area were compared to reveal their indicative characteristics.

## 2. Materials and Method

### 2.1. Study Area and Sampling

The Shewushan gold deposit (N 29°53′31″ and E 113°45′37″) is located in the Wuhan province in the middle of China. The Au content in the strata of the mining area is generally greater than 100 ppb. Strongly weathered siliceous rocks and gravel lie in the topsoil of the profile in the Shewushan mining area. The middle part of the profile is reticulated laterite, the bottom part is brown clay without reticulate structure, and there is brown clay at the base. Gold mainly exists in the brown clay layer in the form of adsorption, and the content of the gold gradually increases downward from the surface, forming ore bodies between the lateritic reticulated clay and brown clay layers where there is enriched clay mineral and limonite [11]. The concentrations of gold and other metals were closely related to the clay mineral, which proved that water in soil was an important factor for the mineralization of gold.

Water stored in soil not only provides the right condition for plants growth. Plant growth also depends on the geochemistry environment for metals accumulation. The dominant plants in the mining area are mainly *Populus canadensis* and *Cinnamomun camphora*.

The collection of soil and plant samples was completed in July. Two typical mine profiles were selected in the Shewushan gold deposit area to carry out continuous sampling (Figure 1, Appendix A and Appendix A). The two soil profiles were divided into the topsoil layer, the lateritic layer, and the clay layer, with deep ore bearing rock from top to bottom according to the profile distribution of Shewushan gold deposit (Figure 2). The location of the sampling profiles are shown in Figure 1. A total of 26 groups of soil samples at different layers in two ore-layer profiles were collected, and seven groups of soil samples were collected in the background profile. In addition, the dominant plants including *Populus canadensis* and *Cinnamomun camphora* were determined by comparing species coverage. The soil samples were collected nearly every other meter from the topsoil layer to the base of the profiles. The plants were collected on the surface of the profiles. For each plant sample, we collected 200 g of fresh leaves. Each soil and plant samples were stored by different numbered bags.

### 2.2. Samples Pretreatment

After soil samples were collected, they were dried in an oven at 60 °C, pulverized, and sifted through a 100-mesh screen. Then, 200 g of the soil samples were weighed for the determination of 40 elements. A prepared sample is digested with perchloric, nitric, and hydrofluoric acids. The residue is leached with dilute hydrochloric acid and diluted to the desired volume. It is then analyzed by inductively coupled plasma-atomic emission spectrometry and inductively coupled plasma-mass spectrometry. The results are corrected for spectral interelement interferences (Appendix A).

For the samples of plants, they were cleaned with deionized water and then dried in an oven at 60 °C and triturated in a ball mill. The powder of the leaves and roots was sifited through an 80 mesh screen (Appendix A). The content of gold (Au) in soil was determined by the fire assay method and atomic absorption spectroscopy (AAS). A prepared sample is fused with a mixture of lead oxide, sodium carbonate, borax, silica and other reagents as required, inquarted with of gold-free silver, and then cupelled to yield a precious metal bead. The bead is digested in dilute nitric acid in the microwave oven, concentrated hydrochloric acid is then added and the bead is further digested in the microwave at a lower power setting. The digested solution is cooled, diluted with de-mineralized water, and analyzed by atomic absorption spectroscopy against matrix-matched standards (Appendix A).

The other elements are analyzed via ICP-MS instrumentation utilizing collision/reaction cell technologies to provide the lowest detection limits available. A prepped sample is cold digested with nitric acid for 8 h before being transferred to hot block for 15 min at 85 °C followed by 2 h at 115 °C. The samples are subsequently cooled and brought up to volume with HCl. The resulting solution is mixed thoroughly and analyzed by ICP-MS and ICP-AES corrected for spectral interferences.

The elements content of soil and plant samples were measured by ALS Minerals-ALS Chemex in Guangdong province, China.

### 2.3. Analysis of Physiological Parameter in Soil and Plants

Determination of chlorophyll (Chl) content in plants: Cut 0.02 g fresh leaves in a mortar and then grind them up with 95% ethanol. The absorbance was determined by spectrophotometer at wavelengths of 665 nm and 649 nm. The concentrations of chlorophyll a and chlorophyll b are calculated as follows [14]:*C_a_* = 13.95*A*_665_ − 6.88*A*_649_;
*C_b_* = 24.96*A*_649_ − 7.32*A*_665_; 
where *C_a_*, and *C_br_* are the concentrations of chlorophyll a and chlorophyll b (mg/g), respectively; and *A*_665_ and *A*_649_ are absorbance at a wavelength of 665 nm and 649 nm.

The MDA content was measured by TBA methods. Fresh leaves samples (80 mg) were homogenized in 4 mL of 0.5% TBA in 20% trichloroacetic acid (TCA). The mixture was heated to 95 °C and then quickly cooled. After centrifugation, the absorbance of the supernatant was read at 532 nm and correction for unspecific turbidity involved subtracting the absorbance at 600 nm. The concentration of MDA was calculated by an extinction coefficient of 155/mm/cm [15]. All samples were performed in duplicate, and the mean values were calculated for analysis. The value of the supernatant MDA concentration was calculated by follow equation.
*C*(μmol/L) = 6.45(*D*_532_ − *D*_600_) − 0.56*D*_450_


The enzyme activity was determined by spectrophotometric method as described by Cho U-H, Zhou W J et al., and Dhindsa R S et al. [14,16,17]: The result is calculated as follows [18]:*SOD* (u/g) = (*A_ck_* − *A_E_*)∗*V*/0.5/*A_ck_*/*V_t_*/*W*

*W* is the fresh weight of leaves, *A_ck_* is the absorbance of the control group; *A_E_* is the absorbance of the sample tube; *V* is the total volume of sample solution; and *V_t_* is the amount of enzyme liquid used in the determination.

The increase of POD value by 0.01 per minute was defined as 1 enzyme activity unit, and it can be calculated by following [19]:*POD* (u/g) = (Δ*A*_470_ × *V*)/(*W* ×*V_s_* × 0.01 × *t*)
where Δ*A*_470_ as reaction time (the change of absorbance); *V* is the total volume of sample solution; *W* is the fresh leaf weight of the sample; *V_s_* is the liquid volume of enzyme used for determination; and *T* is the reaction time.

## 3. Result and Discussion

### 3.1. Geochemical Characteristics of Soil Profile in Mining Area

The horizon differentiation of the layers in the soil profile is the result of natural soil formation and long-term human activities. The soil profile in the Shewushan gold deposit area records the migration and accumulation process and vertical distribution characteristics of gold and other metallogenic elements. With the influence of long-term weathering, tectonic activity, and mining activity, the types and contents of soil metal elements at different depths in the mining area were significantly different from the soil profile in the background area. As can be seen from the Figure 3, the background profile of both of two mine profiles had similar stratified soil layers. The top soil with humus layer is on the surface at the depth of 2.0 m–2.4 m, the middle part is laterite, and the bottom part is brown clay where gold mainly concentrated in. The soils geochemical characteristics of the three profiles were shown in Table 1 and Table 2. The content of ore elements in the profile and the background profile are shown in Figure 3 and Table 1 and Table 2. The average content of Ca, Mg, and K in the soil of the mining area were 1.4%, 1.24%, and 2.5%, while the average content of Fe and Al were 4.87% and 8.05% respectively.

From the Table 1 and Table 2, the content of Fe in the laterite and brown clay layer was greater than in the topsoil, and gradually decreased with the increasing depth in the profile. The average content of Ca, Mg, and K in the soil in the background area is 0.18%, 0.135%, and 0.475%, and the content of Fe and Al were 4.37% and 7.66%, respectively. The contents of ore elements at depth were lower than those at top. The contents of Ca, K, and Mg in the soil in the mining area were significantly lower than that in the background profile, while the contents of Fe and Al were slightly higher than that in the background profile, respectively.

It can be seen from Figure 4 that the gold was mainly located at a depth of 7–11 m in the soil profile of the mining area. Most of the gold was concentrated in the lateritic layer and clay layer, and the contents of Fe in the 1–3 m topsoil were significantly higher than those in the background profile. In the process of soil-forming, potassium, sodium, calcium, magnesium, and iron were released under oxidizing stress. The average content of Ca, Mg, K and Na in the surface soil in background profile is 12.86%, 10.89%, 18.96%, and 6.67% compared to the mine area. These data indicate the weathering process. Gold can be enriched by clay mineral and limonite in the oxidizing environment in topsoil and may be absorbed by some dominant plant species in mine. The contents of Mg, Fe, Au, Pb, Cu, Mg, Mn, and Ag varied greatly in each layer of the profile, showing the enrichment and differentiation characteristics of trace elements from the deep layer of the soil to the topsoil layer. The contents of the heavy metal elements Ag, Sn, Au, Hg, Pb, Cu, and As in the topsoil of mine profiles were 7.0, 5.2, 4.7, 4.5, 4.5, 4.2, 2.6, 2.3, and 2.1 times of the background values of Chinese soil, respectively.

The contents of metals in Figure 3 were unified by normalization processing. Compared to the background, gold could be detected in all layers with high levels of Pb. Except for silver, the concentrations of gold (Au), plumbum (Pb), arsenic (As), antimony (Sb), iron (Fe), and cobalt (Co) in mine profiles were much higher than that in background profile, especially in the laterite and the brown clay layer (which may be the effect of gold mineralization and weathering). From Table 2 and Table 3, it can be seen that the contents of As and Sb in the brown clay soils were as much as 2.7 times and 30 times larger in the mine profiles than in the background profile, while the content of Co in the topsoil was 7 times larger than background profiles.

### 3.2. Enrichment Characteristics of Trace Elements in Soil by Plants

Compared to the topsoil of the background profile, Au, Co, As, and Pb with high contents should be focused upon, as dominant plants may accumulate these elements from the topsoil.

The contents of heavy metals in plants are not only related to the mobility of metal elements but are also closely related to plants’ tolerance to metals stresses [8,20]. Elements in the soil are the main source of elements in plants, and therefore the elements related to mineralization will be enriched in the dominant plants. Compared with the plants growing in background profile, dominant species in the mine area are better adapted to the mine soil conditions [21,22,23]. The dominant plants have the capacity to enrich ore-forming elements enrichment and some may be sensitive to the mine environment. The parameters and indexes of these species could be indicators for locating mineral resources. The biological absorption coefficient (BAC) can indicate the extent to which elements are transferred from the soil to the plant. The BAC can be calculated as follows [24]:BAC = *P_x_*/*S_x_* (soil) 
where *P_x_* represents the content of the element in the plant and *S_x_* represents the content of the element in the root soil.

As can be seen in Table 3, the biological absorption coefficients of trace elements in different plants under the same growth environment were different. The Mn, Zn, Ba, Cd, and Pb biological absorption coefficient of the *Populus canadensis* and camphor were greater than 0.1. Plants enrichment of trace elements from the soil does not completely depend on certain element content in the soil; plants can exist and adjust the physiological mechanism of trace element levels in their parts. Therefore, it is necessary to evaluate the ability of plant leaves to indicate soil geochemical anomalies by integrating the contrast coefficient of each element.

The contrast coefficient (CM) is calculated by the equation:CM = *C_mine_*/*C_background_*.
where *C_mine_* is the average value of an element in the species in the mine area. *C_background_* is the average element content values of plants in background region.

This coefficient can assess the concentrating ability and sensitivity of plants. According to the coefficient, the contrast of elements in plants can be divided into four levels: low (CM < 3), medium (3 ≤ CM < 5), high (5 ≤ CM < 10), and very high (10 ≤ CM). The larger the CM of an element, the more obvious the intensity indicator of target elements [25]. Table 4 proved that Au, Ag, Pb, K, Cu, Co, As in the mine area are all higher than the background area. The elements with high contrast coefficient in *Cinnamomun camphora* were Au (74.6) and Co (7.78). The contrast coefficients of Au, Ag, Co and As in *Populus canadensis* leaves were 34.5, 2.53, 4.06 and 6.03, respectively. The contrast coefficient shows that the content of Au, Co and As can be indicated by *Populus canadensis* and *Cinnamomun camphora*.

### 3.3. Effects of Trace Elements on Biochemical Characteristics of Indicator Plants

The absorption of heavy metal elements in plants will cause a series of biological reactions, which can be observed as changes in the enzyme activity and malondialdehyde (MDA) content [26,27]. Toxic elements can produce free radicals that lead to membrane lipid peroxides and cell membrane damage [28]. Therefore, MDA content can reflect the degrees of the toxic elements stress. The content of MDA in plant leaves in mine profile was greater than that growing in background profiles (Table 5 and Appendix A). With the increase of content of Au, Cu, Zn, Hg, the MDA content in *Populus canadensis* leaves increase gradually.

The chlorophyll contents of Populus canadensis and Cinnamomun camphora with various concentrations of Au, Cu, and Zn were shown in Table 6. Compared to the background profile, the chlorophyll content of poplar and camphor growing in the mine profiles was reduced by 19.2% and 21.6%, respectively, which means that the chlorophyll content of the two species was sensitive to these metals. Plants have an anti-oxidation enzyme system to mop up free radicals [18]. Superoxide dismutase (SOD) and peroxidase (POD) are clearance agents for free radicals which can clear lipid peroxides and have roles to play in protecting cells and tissue; they can work against the oxidative damage caused by free radicals on the plant cell membrane structure [25,29].

Figure 5 proved that the POD and SOD activities in the canadensis grown in background profiles were 2.6 times and 2.1 times lower than those in mine profiles, respectively. The SOD and POD contents in camphor leaves were 1.5 and 1.8 times greater than those in the background profiles, respectively, which reflected that the protective enzyme system in leaves of the two plants was activated. In general, SOD and POD showed an increasing trend with the increase of concentrations of Cu, Au, and Pb. Therefore, the *Cinnamomun camphora* and *Populus canadensis* can grow normally under the stress of Au, As, Pb, and Co.

## 4. Conclusions

(1) The contrast coefficient of Au and As in *Populus canadensis* growing in gold deposit were 34.5 and 6.03, while the values of Au and Co in *Cinnamomun camphora* were 74.6 and 7.78. These results show that the two species are sensitive to the Au and could be regarded as indicators of gold accumulation in mines.

(2) Compared to the samples in background profiles, the contents of chlorophyll and MDA experienced no obviously changes in leaves of *Cinnamomun camphora*. The activity of SOD and POD in the leaves of *Populus canadensis* and *Cinnamomun camphora* are sensible to the contents of metals. All of the results indicate that *Populus canadensis* and *Cinnamomun camphora* can enrich Au, Co, and As in the gold mining area, and that they have good tolerance towards the metals. The content of gold and associated elements in leaves of *Populus canadensis* and *Cinnamomun camphora* reflects the soil geochemical environment at the depth 5–11 m depth, which is indicative of a need for thorough investigation in this area.

## Figures and Tables

**Figure 1 plants-11-00038-f001:**
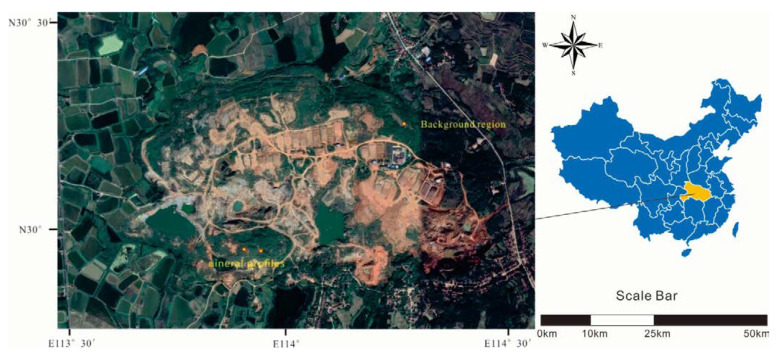
The sampling area in Shewushan gold deposit.

**Figure 2 plants-11-00038-f002:**
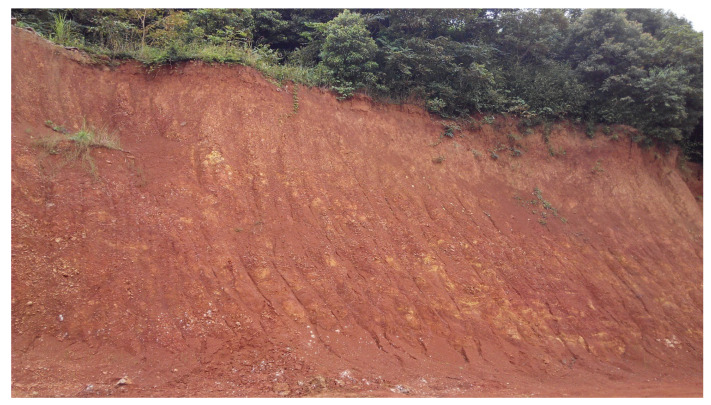
The sampling profiles in Shewushan gold deposit.

**Figure 3 plants-11-00038-f003:**
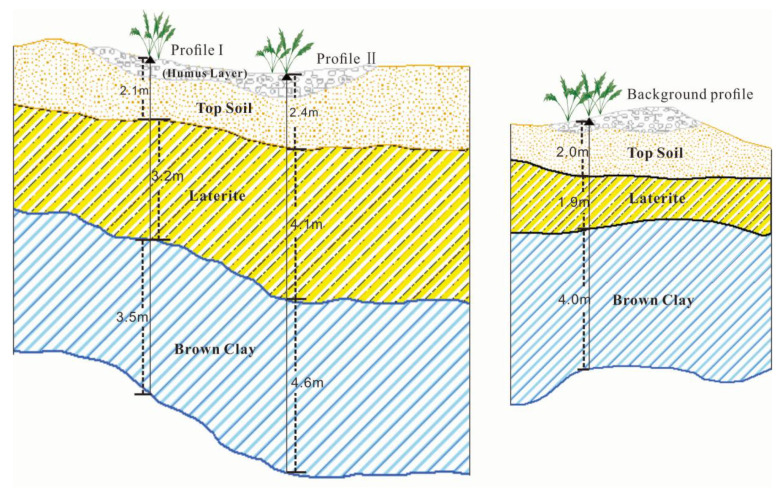
The stratified soil layers of sampling profiles.

**Figure 4 plants-11-00038-f004:**
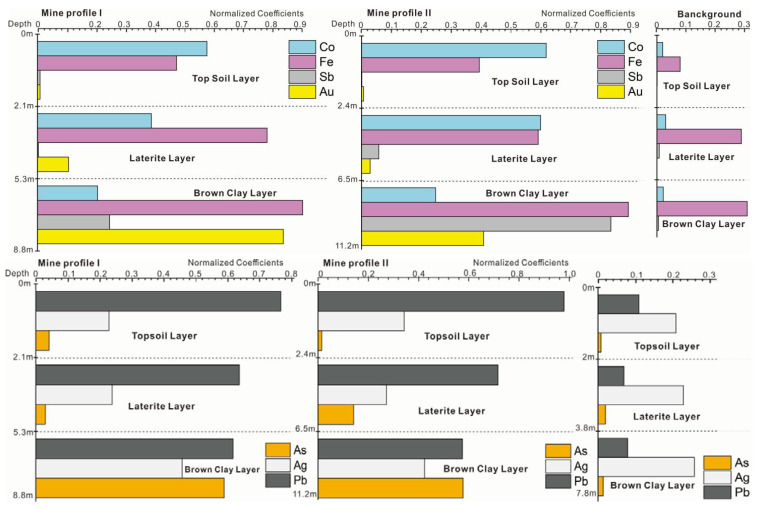
The content of each ore-forming elements in ore profile.

**Figure 5 plants-11-00038-f005:**
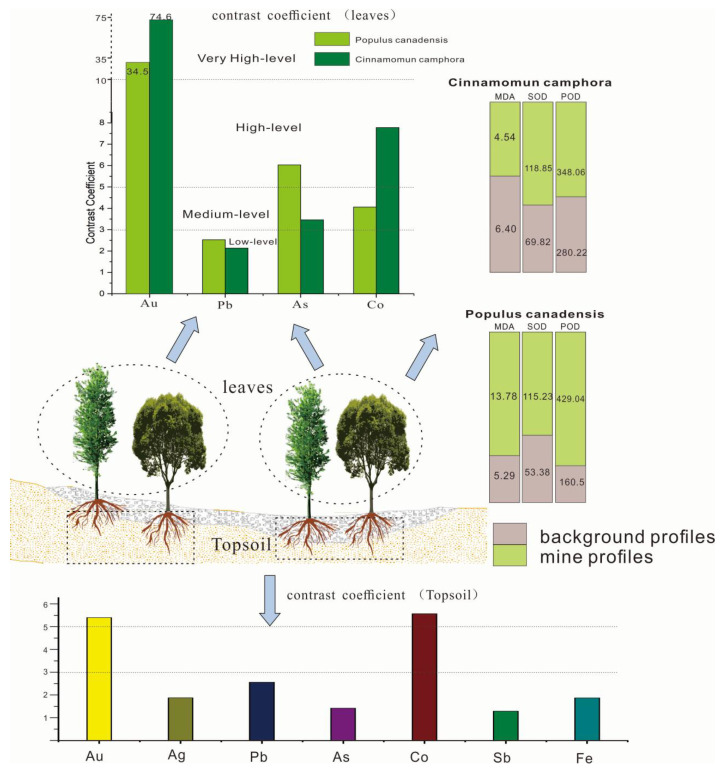
The contrast coefficient, MDA content, and enzymatic activity of plants in mine and background profiles.

**Table 1 plants-11-00038-t001:** The elements concentration of mine profiles.

Profile I	Layer	Au	Ag	Pb	As	Sb	Fe (%)	Co
Depth (m)		μg/g	mg/kg					
0.7	Top soil	0.018	0.07	37.7	36.6	3.7	3.87	20.6
1.4		0.010	0.03	34.1	22.8	2.24	4.32	26.2
2.1		0.012	0.02	35.3	30.9	2.78	3.65	21.3
2.8	Laterite	0.009	0.04	32.4	25.4	2.43	4.82	22.3
3.5		0.028	0.05	32.6	24.7	2.32	4.87	16.2
4.5		0.077	0.02	31.6	24.3	2.24	5.14	15.5
5.3		0.115	0.06	30.3	28.2	2.06	6.13	11.4
5.6	Brown clay	0.161	0.08	25.9	27.5	2.86	5.89	9.5
6.6		0.433	0.06	30	75.7	4.68	5.77	7.5
7.6		0.613	0.17	28	190.6	33.7	5.29	4.3
8.2		0.937	0.11	36.2	299.8	45.1	5.96	5.4
8.8		0.876	0.14	35.7	314.5	41.3	5.92	5.8
Profile II	layer	Au	Ag	Pb	As	Sb	Fe (%)	Co
Depth (m)		μg/g	mg/kg					
0.8	Top soil	0.005	0.09	41	25	2.3	3.66	21.2
1.5		0.015	0.08	42.2	24.6	1.97	4.36	24.3
2.4		0.022	0.09	42.7	18.1	1.83	4.95	23.6
2.8	Laterite	0.008	0.05	36.8	18.4	1.83	4.25	34.7
3.5		0.006	0.03	39.6	17.9	1.79	4.15	44.8
4.2		0.008	0.04	38	20.4	1.96	4.37	31.3
4.9		0.017	0.04	21	15.4	3.76	3.58	7
5.6		0.015	0.08	30.8	16.3	4.72	3.77	40.3
6.5		0.026	0.05	27.1	14.4	6.53	3.48	13.3
7	Brown clay	0.025	0.03	34.5	37.9	20.9	4.53	16.5
7.7		0.037	0.02	31.2	83.5	10.89	3.9	21.4
8.4		0.055	0.02	19.7	170.1	52.15	4.73	23
9.1		0.105	0.01	33.2	137.6	74	5.39	6.2
9.8		0.252	0.11	16.6	174.3	94.1	6.62	5.1
10.5		0.490	0.22	14.3	361.2	130	4.78	8.1
11.2		0.763	0.19	21.2	258.6	122	6.17	4.6

**Table 2 plants-11-00038-t002:** The elements concentration of background profiles.

Background Profile	Layer	Au	Ag	Pb	As	Sb	Fe (%)	Co
m		μg/g	mg/kg					
1	Top soil	<0.005	0.08	20.1	17.5	1.9	2.78	3.3
1.5		<0.005	0.09	13.2	18.3	2.08	1.85	4.7
2		<0.005	0.07	11.8	19.4	1.76	1.98	4.3
3	Laterite	<0.005	0.05	13.1	21.2	2.23	2.46	3.9
3.8		<0.005	0.02	15.4	23.8	2.64	3.52	4.2
5	Brown clay	<0.005	0.11	13.4	24.5	3.51	2.85	5.1
6.5		<0.005	0.09	12.4	22.8	2.29	3.44	4.3
7		<0.005	0.06	16.9	20.6	2.88	2.97	3.6
7.8		<0.005	0.07	14.2	17.4	2.14	3.71	4.6

**Table 3 plants-11-00038-t003:** Biological absorption coefficient of dominant plants.

Elements	Au (SD)	Zn (SD)	Cu (SD)	Co (SD)	As (SD)	Fe (SD)	Al (SD)
Populus canadensis	0.08 (0.02)	1.37 (0.34)	0.33 (0.07)	0.25 (0.03)	0.03 (0.01)	0.01 (0.01)	0.01 (0.01)
Cinnamomun camphora	0.04 (0.01)	0.37 (0.25)	0.25 (0.08)	0.01 (0.01)	0.21 (0.03)	0.01 (0.01)	0.01 (0.01)

**Table 4 plants-11-00038-t004:** Elements content and contrast coefficient of dominant plants (mg/kg).

Species	Elements	Background (Average)	SD	Mine Profile (Average)	Contrast Coefficient
Populus canadensis	Au	0.002	0.08	0.069	34.50
	Ag	0.015	0.01	0.038	2.53
	Pb	1.97	2.69	3.03	1.54
	Ca	2.36	1.48	2.50	1.06
	K	2.77	1.05	4.76	1.72
	Mg	1.88	0.39	3.35	1.78
	Zn	22.24	0.44	15.91	0.72
	Cu	5.23	0.72	8.11	1.55
	Co	0.87	0.56	3.53	4.06
	As	0.72	1.48	4.34	6.03
	Fe	5.67	3.3	6.58	1.16
	Al	3.42	2.57	4.34	1.27
	S	2.45	1.24	2.01	0.82
**Species**	**Elements**	**Background Values**	**SD**	**Mine Profile (Average)**	**Contrast Coefficient**
Cinnamomun camphora	Au	0.006	0.14	0.448	74.60
	Ag	0.011	0.01	0.029	2.64
	Pb	1.89	1.05	4.04	2.14
	Ca	1.36	4.32	1.43	1.05
	K	2.67	2.09	5.69	2.13
	Mg	3.88	1.35	2.64	0.68
	Zn	17.52	3.33	20.71	1.18
	Cu	5.12	0.87	8.03	1.57
	Co	0.23	0.64	1.79	7.78
	As	1.52	1.32	5.26	3.46
	Fe	7.67	1.81	7.82	1.02
	Al	2.32	2.14	2.06	0.89
	S	2.05	1.65	2.69	1.31

**Table 5 plants-11-00038-t005:** The enzyme content of dominant plants.

Sampling Area	Species	MDA (SD)	SOD (SD)	POD (SD)
Mine soil profile	Populus canadensis	13.78 (3.21)	115.23 (30.67)	429.04 (83.34)
	Cinnamomun camphora	4.54 (1.05)	118.85 (27.97)	348.06 (75.79)
Background region	Cinnamomun camphora	6.4 (1.29)	69.82 (10.93)	280.22 (58.27)
	Populus canadensis	5.29 (0.89)	53.38 (11.58)	160.5 (21.94)

**Table 6 plants-11-00038-t006:** The contents of cholophyll and metals.

Sample Location	Species (Sample Size)	Cholophyll a + b (SD)	Au (SD)	Co (SD)	As (SD)
		mg/g	μg/g	mg/kg	
Mine profile	Populus canadensis (17)	1.91 (0.42)	0.069 (0.08)	3.53 (0.56)	4.34 (1.48)
	Cinnamomun camphora (14)	1.38 (0.27)	0.448 (0.14)	1.79 (0.64)	5.26 (1.32)
Background profile	Populus canadensis (5)	2.88 (0.72)	0.002 (-)	0.87 (0.22)	0.72 (0.26)
	Cinnamomun camphora (4)	2.03 (0.76)	0.006 (0.005)	0.23 (0.09)	1.52 (0.34)

## Data Availability

Measured data and photos of sample area could be found in Appendix A.

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
