# Peer review of "Biogeochemistry of Dominant Plants and Soils in Shewushan Gold Lateritic Deposit, China"

_plants, 2021, doi:10.3390/plants11010038_

Round 1

Reviewer 1 Report

In the present manuscript, the authors try to describe, explain and interpret the accumulation of metals and, in particular, gold (Au) by three plant species (Populus x Canadensis, Cinnamomum camphora and Rhus chinensis) grown in soils of a gold mining area in the Shewushan gold deposit, China. The authors try to report metal concentrations in different soil layers of the deposit and in plant roots and leaves, and to relate the concentrations found to the gold content of the deposit. In addition, they measure the chlorophyll content and some enzyme activities (Malondialdehyde, SOD, and Peroxidase) in plant tissues and try to relate these values to a purposed tolerance of plants towards gold and other trace elements.

I must say that it was rather disappointed in reading this paper. In fact, the manuscript is largely deficient an neglects an accurate description of the locality. For example, there is no indication of about what the so-called “background” region should be, and how this was compared to the gold field area. The authors also miss to tell the reader in which province of China the supposed gold field may be located. Moreover, they speak about different soil layers, but there is no description of these layers at all (except of a photography of the soil profile). What is layer A, what are the characteristics of the other layers mentioned in the text? In addition, there are serious deficiencies in the description of methods. No details about measurement of metal concentrations or preparation of samples are given. How exactly were metal concentrations assessed, were the values obtained compared with those of standardized soil reference material, etc. etc.? Also, the assessment of chlorophyll and of enzyme activity measurements lack proper methodical references. A clear explanation of “absorption coefficients” (for plants) and of “contrast coefficients” for soil samples are missing. In the Results section, the authors report concentrations of metal distribution in the soil (Figure 3), but there is no mention or indication about “layer A” mentioned in the text. Moreover, there is no comparison between metal contents in soil of the gold mining area with those of the so-called “background region”, and nobody knows what the authors mean by “background region”. In Table 2, the authors report metal values but a clear description of what these values mean is missing. According to the author’s explanation, the contrast coefficient should apply to soil samples (element concentrations in “background” soil versus gold field soil). In spite of this, they speak about “contrast coefficients” in plants (instead of soil). In Figure 4, the hatching pattern of the bars shown is inconsistent and unclear compared to the legend. In Table 3, they compare enzyme activities in plant leaves from the mining area with those of plants from a not-defined background region. Altogether, a very confusing and bad manuscript, which is also deficient from a language point of view.

I’m sorry that in consideration of all these lacks and deficiencies, I can only suggest to refuse this manuscript.

Author Response

Thank you for your comments concerning our manuscript entitled “Biochemistry of dominant plants and soils in Shewushan gold lateritic deposit, China (plants-1462404). Those comments are valuable and very helpful. We have read through comments carefully and have made corrections.

Please see the attachment for the details.

Reviewer 2 Report

The work presented by Qin and Wang investigated the chemical profile of soils collected from a mining area in China as well as their concentration in plants (P. canadensis, C. camphora and R. chinensis) growing inside it, thus evaluating also their physiological status. A comparison with data collected from soils and plants from background areas was also progressed. Overall, this work is very interesting and well-written but, although that, some issues need to be checked:

Major issues

Line 77: Did the Authors mix all leaves without preserving their origin? If yes, the Authors lost data variability. Is this the motivation for which all graphs and tables lack indicators of data distribution (standard error or standard deviation)? Please specify the method. Was this method also applied to soil and leaf samples for physiological analysis? If the Authors conserved the replicates, please add data variability indicators for each conducted measurement. 

Line 82: Leaves collection was progressed following a particular method? At which height leaves were collected and in which season (month)? Please add some details.  

Line 85: add a special paragraph for this section. Reference the used method and add information on quality control (recovery and precision of the measurements).

Line 221: Are your findings coherent with those of other studies? Add some examples.

Minor issues:

Line 55: remove this phrase and add the coordinate in the next line. "The Shewushan gold deposit (N29°53 '31 ", E113°45' 37") is located".

Line 114: check repetitions

Author Response

Thank you for your comments concerning our manuscript entitled “Biochemistry of dominant plants and soils in Shewushan gold lateritic deposit, China (plants-1462404). 

Please see the attachment for details.

Reviewer 3 Report

General Comments
The structure of the manuscript is not as per the instruction provided by the journal (https://www.mdpi.com/journal/plants/instructions#manuscript ). The manuscript quality is poor and, the study done is lacking the presentation clarity. The author is suggested to clarify the major objectives and discus the gap area in a convincing manner. Furthermore, efforts on the manuscript should be given to include some more studies on the mineralization and dynamics of minerals in soil plant system.

This manuscript should be considered only after a major revision.  However, a few basic comments for the authors worth-noting could be listed as follows:

Major:

  • A number of factors were discussed in the manuscript regarding effects of minerals on plants. Any data/suitable explanation on dynamics of minerals with water retention capacity of the soil (affected due to these minerals) could be an interesting addition.
  • Image quality of the graph is poor, and it has no error bars, however, standard deviation is shown on the table. If graph was presented well and convincing, then table would be needed to include
  • All the scientific names must be italicized

Minor:

  • Rigorous work should be done on abbreviations, spacing, punctuation, fonts, typo etc. in the entire manuscript
  • Font discrepancy in the heading ‘Materials and Methods’
  • Ln 23: ‘lipid peroxide’; spelling error
  • Ln 84: ‘sifted’; spelling error
  • Ln 87-88: some parts have been italicized for no reason
  • Font discrepancy in the heading ‘Results and discussion’
  • Ln 153: ‘kalium’; English name must be used instead of Latin name
  • Fig 4: caption: ‘plants’; spelling error
  • Ln 245-245: some parts have been italicized for no reason
  • Difference in font of Table 3 from the rest of manuscript
  • Use abbreviations correctly and uniformly
  • All the schemes should be numbered in the Roman number
  • Minutes had been mentioned as “min.” or “minutes”

Author Response

(The authors gave the same response as above.)

Reviewer 4 Report

The manuscript deals with a relevant subject to PLANTS, but shows some important points presented below to which the authors should address. Moreover, a deeper discussion is needed and the English should be strongly revised in order to make the manuscript more readable and fluent. Thus, I recommend that the manuscript, in the present form, should be accepted after major revision.

Specific points:

  1. Lines 14-15: reformulate sentence.
  2. Lines 16-18: reformulate sentence.
  3. Line 20: tolerated species? Change.
  4. Lines - 24- 27: References should be included.
  5. Authors should reformulate sentences (lines 24-29).
  6. Reformulate line 55.
  7. Authors should include reference to Figure 1 in section 2.1 (between lines 51-66).
  8. Lines 87-88: Authors should reformulate the sentence. The same for lines 96-98 and lines 108-112.
  9. Line 89: Where was used the absorbance at 470 nm.
  10. Statistical analysis is missing in materials and methods section.
  11. Line 140: Table 1 presents Biological absorption coefficients of dominant plants. and not the content of elements in the profile and the background.
  12. Line 226: figure 2?
  13. Lines 230-231: remove (out context in this form).
  14. Where are the chlorophyll results?

Author Response

(The authors gave the same response as above.)

Round 2

Reviewer 1 Report

Dear authors, you have answered sufficiently clear to my critical review, providing now an extensively corrected and improved manuscript, with a lot of more significant information added to the mansucript and in your supplementary files. Congratulations.

Reviewer 2 Report

The authors significantly improve the manuscript according to the proposed indications. I recommend authors to check only the following:

Line 260: plants

Table 5 and 6: The values are the averages? For Tab 5, put the name of the species in the same order.

Line 282: check "that the have"

Reviewer 3 Report

Biogeochemistry of dominant plants and soils in Shewushan 3 gold lateritic deposit, China 

General Comments

The author has presented a detailed and well-collated study on the Biogeochemistry of dominant plants and gold deposit soils of Shewushan. The author has elaborated the materials and methods giving a clear insight into his research. Revision done by the author is appreciated. 

Major comments:

  • The author did not change the structure of the manuscript as per the instruction provided by the journal https://www.mdpi.com/journal/plants/instructions#manuscript
  • The author was given some significant comments (in the first review) to improve the quality of the manuscript in terms of typos and grammatical errors, the author has failed to address this in the manuscript.
  • The author is suggested to read the entire manuscript carefully before communicating to the journal.

Minor comments:

  • All the scientific names must be italicized.

Though this comment was given in the first review, the author has done some corrections and some names are yet to be italicized (table 6)

  • Representation of units should be uniform across the manuscript
  • With standard deviation values provided in the tables, can the authors add error bars for the graphs provided in the manuscript?
  • Spacing and punctuation errors in the entire manuscript
  • Ln 282: ‘…that they have..’; incorrect sentence formation
  • Ln 285: ‘thorough’; spelling error
  • Ln 72: (FIg2) typo error
  • Similar to above-mentioned typo, punctuation and uniformity has to be checked throughout the manuscript

Reviewer 4 Report

 I have read the revised version of the manuscript. The manuscript has been 
improved to warrant publication in Plants.